# Targeting mGlu1 Receptors in the Treatment of Motor and Cognitive Dysfunctions in Mice Modeling Type 1 Spinocerebellar Ataxia

**DOI:** 10.3390/cells11233916

**Published:** 2022-12-03

**Authors:** Francesca Liberatore, Nico Antenucci, Daniel Tortolani, Giada Mascio, Federico Fanti, Manuel Sergi, Giuseppe Battaglia, Valeria Bruno, Ferdinando Nicoletti, Mauro Maccarrone, Serena Notartomaso

**Affiliations:** 1IRCCS Neuromed, 86077 Pozzilli, Italy; 2Department of Physiology and Pharmacology, Sapienza University, 00185 Rome, Italy; 3Department of Pharmacology and Neuroscience, School of Medicine, Texas Tech University Health Sciences Center, Lubbock, TX 79430, USA; 4European Center for Brain Research (CERC)/Santa Lucia Foundation IRCCS, 00143 Rome, Italy; 5Faculty of Bioscience and Technology for Food Agriculture and Environment, University of Teramo, 64100 Teramo, Italy; 6Department of Chemistry, Sapienza University, 00185 Rome, Italy; 7Department of Biotechnological and Applied Clinical Sciences, University of L’Aquila, 67100 L’Aquila, Italy

**Keywords:** SCA1, mGlu1 receptor, endocannabinoids, hippocampus, learning and memory

## Abstract

Type 1 spinocerebellar ataxia (SCA1) is a progressive neurodegenerative disorder with no effective treatment to date. Using mice modeling SCA1, it has been demonstrated that a drug that amplifies mGlu1 receptor activation (mGlu1 receptor PAM, Ro0711401) improves motor coordination without the development of tolerance when cerebellar dysfunction manifests (i.e., in 30-week-old heterozygous ataxin-1 [154Q/2Q] transgenic mice). SCA1 is also associated with cognitive dysfunction, which may precede cerebellar motor signs. Here, we report that otherwise healthy, 8-week-old SCA1 mice showed a defect in spatial learning and memory associated with reduced protein levels of mGlu1α receptors, the GluN2B subunit of NMDA receptors, and cannabinoid CB1 receptors in the hippocampus. Systemic treatment with Ro0711401 (10 mg/kg, s.c.) partially corrected the learning deficit in the Morris water maze and restored memory retention in the SCA1 mice model. This treatment also enhanced hippocampal levels of the endocannabinoid, anandamide, without changing the levels of 2-arachidonylglycerol. These findings suggest that mGlu1 receptor PAMs may be beneficial in the treatment of motor and nonmotor signs associated with SCA1 and encourage further studies in animal models of SCA1 and other types of SCAs.

## 1. Introduction

Spinocerebellar ataxias (SCAs) are a heterogeneous group of genetic disorders characterized by progressive impairment of gait and motor coordination. While cerebellar degeneration is the hallmark of SCAs, other CNS regions may be affected, and their dysfunction may precede the onset of motor signs [1]. SCA1 accounts for 6 to 27 percent of all autosomal-dominant cerebellar ataxias, with a prevalence of 6–27/100,000, according to the estimates of the US National Institute of Health. Prevalence, however, is higher in central Poland and eastern Siberia [2,3]. Typical signs and symptoms of SCA1 are ataxic gait, speech and swallowing difficulties, muscle stiffness, weakness of ocular muscles, and cognitive dysfunction with impairment of recent and remote memory [4,5]. Patients affected by SCA1 typically survive 10 to 20 years after clinical onset, and no disease-modifying or symptomatic drugs are currently available. SCA1 is caused by an expansion of a CAG repeat in the ataxin-1 encoding gene [6]. The resulting polyglutamine expansion prevents the interaction between ataxin-1 and retinoid-related orphan receptor α (RORα), a transcription factor that is highly expressed in cerebellar Purkinje cells. This impairs the expression of RORα-regulated genes in Purkinje cells and other RORα-expressing cells [7]. A functional link between RORα and type-1 metabotropic glutamate (mGlu1) receptors is suggested by the evidence that mice expressing ataxin-3 with an expanding polyglutamine stretch show reductions in both RORα and mGlu1 receptor signaling proteins, and pharmacological activation of RORα corrects morphological, functional, and behavioral deficits in these mice [8]. In Purkinje cells, mGlu1 receptors are highly expressed and play a key role in the maturation of the cerebellar cortex and in mechanisms of the activity-dependent synaptic plasticity underlying motor learning [9,10,11]. During postnatal development, activation of mGlu1 receptors drives the second phase of elimination of supernumerary climbing fibers innervating Purkinje cells [12,13] and allows for the segregation of climbing and parallel fibers in different territories of Purkinje cell dendrites [14]. mGlu1 receptors are essential for the induction of long-term depression (LTD) at the synapses between parallel fibers and Purkinje cell dendrites, and the lack of mGlu1 receptors in Purkinje cells results in a severe impairment of motor coordination and conditioned eyeblink reflex [15,16,17,18,19]. In humans, neutralizing anti-mGlu1 receptor antibodies have been associated with paraneoplastic ataxia in patients with Hodgkin’s lymphoma [20,21,22]. mGlu1 receptors are coupled to G_q/11_, and their activation stimulates phospholipase-Cβ, the enzyme that converts phosphatidylinositol-4,5-bisphosphate into inositol-1,4,5-trisphospate (InsP3) and diacylglycerol (DAG) [23]. Abnormalities in the expression of mGlu1 receptors and/or mGlu1 receptor signaling in Purkinje cells have been reported in mice modelling different types of SCAs [24,25]. Here, we focus on SCA1. In the first part, we critically review the existing evidence linking mGlu1 receptors in Purkinje cells to the pathophysiology of cerebellar dysfunction in the SCA1 mice model. In the second part, we extend the study of mGlu1 receptors to the hippocampus of SCA1 mice, presenting new data that suggest a role for mGlu1 receptors in cognitive dysfunction associated with SCA1.

### 1.1. mGlu1 Receptors and Cerebellar Dysfunction Associated with SCA1

Studies on mGlu1 receptors in SCA1 mouse models have generated conflicting data. However, most studies demonstrate defective mGlu1 receptor expression and/or signalling in the cerebellar Purkinje cells of the SCA1 mice model. Mice carrying the *staggerer* (sg) mutation recapitulate some of the morphological and behavioural abnormalities typical of SCA1. The *sg* mutation consists of a 122 bp deletion within the region of the RORα gene encoding the ligand-binding domain of the protein [26]. Homozygous *sg/sg* mice showed a complete loss of electrophysiological responses mediated by mGlu1 receptors at the synapses between parallel fibers and Purkinje cell dendrites. In addition, these mice showed reduced expression and mislocalization of mGlu1 receptors in Purkinje cells [27]. Although *sg* mutant mice are not characterized by polyglutamine tract expansion in ataxin-1, they share with a bona fide SCA1 mice model defective transcriptional activity of RORα [28,29]. Defects in mGlu1 receptor-mediated responses (e.g., slow EPSPs, dendritic Ca^2+^ signals, activity-dependent synaptic plasticity at parallel fiber-Purkinje cell synapses) were also observed in transgenic mice carrying the human ataxin-1 gene with an extended 82 CAG repeat under the control of a Purkinje cell-specific promoter. These defects were progressive and already observed at early stages of the disease [30]. The mechanisms underlying the defective mGlu1 receptor signalling in the SCA1 mice model are only partially elucidated. Regulators of G-protein signalling (RGS) form a family of GTPase-activating proteins that negatively modulate G-protein signalling [31]. RGS8, which is specifically expressed in Purkinje cells, is dysfunctional in mouse models of SCA1, SCA2, SCA7, and SCA14. Increases in RGS8 levels might help to restrain mGlu1 receptor signalling in the SCA1 mice model [1].

Using conditional SCA1 [82Q] transgenic mice, Zu et al. have shown that doxycycline-induced suppression of the mutant ataxin-1 gene in the early/mid stages of the disease rescued the pathological phenotype, and this was associated with a reappearance of mGlu1 receptors in the parallel fibre-Purkinje cell synapses in the molecular layer of the cerebellar cortex [32]. We have found a reduced expression of mGlu1α receptors in Purkinje cells of 30-week-old heterozygous ataxin-1 [154Q/2Q] transgenic mice, which, at this age, show a severe impairment in motor coordination [33]. Interestingly, systemic treatment with a selective positive allosteric modulator (PAM) of mGlu1 receptors (compound Ro0711401) caused a long-lasting improvement in motor performance in these mice, which outlasted the clearance of the drug from the cerebellum. The symptomatic benefit persisted after repeated injections of Ro0711401 with no development of tolerance [33]. Treatment with Ro0711401 also enhanced Ser880 phosphorylation of the GluA2 subunit of AMPA receptors, a process that promotes AMPA receptor internalization and induction of LTD [34], and corrected abnormalities of dendritic spines in Purkinje cells of the SCA1 mice model [33]. Taken together, these findings suggest that SCA1 is associated with defective mGlu1 receptor signalling in cerebellar Purkinje cells and that mGlu1 receptor PAMs are putative candidate drugs in the treatment of cerebellar motor symptoms associated with SCA1.

Contrastingly, there was evidence that mGlu1 receptor-dependent synaptic signalling was prolonged in 12-week-old SCA1 [82Q] transgenic mice and that an mGlu1 receptor blockade with the selective negative allosteric modulator (NAM), JNJ16259685, normalized synaptic responses mediated by mGlu1 receptors and improved the ataxic phenotype of these mice [35]. How these findings can be reconciled with all the reported studies above is unclear.

A potential concern related to the use of mGlu1 receptor PAMs in SCA1 is that pharmacological activation of mGlu1 receptors (despite the reduced receptor signalling) may drive mechanisms of excitotoxicity in Purkinje cells as a result of InsP3-intracellular Ca^2+^ release. It will be interesting to examine whether prolonged treatment with Ro0711401 or other mGlu1 receptor PAMs affects Purkinje cell degeneration and survival in a SCA1 mice model.

mGlu5 receptors, which are also coupled to G_q/11_ proteins, are absent in mature Purkinje cells, but they are expressed and functional in Purkinje cells during the first two weeks of a mouse’s postnatal life [36]. The developmental pattern of expression of mGlu1 and mGlu5 receptors is complementary in cerebellar Purkinje cells, and endogenous activation of mGlu1 receptors down-regulates the expression of mGlu5 receptors through an epigenetic mechanism [36]. A recent article published in this SI on Cells demonstrates that transgene-induced expression of mGlu5 receptors in Purkinje cells rescued the pathological phenotype in mGlu1 knock-out mice, suggesting that mGlu1 and Glu5 receptors behave similarly in Purkinje cells [37]. Interestingly, mGlu5 receptors are re-expressed in Purkinje cells of SCA1 mice [33], perhaps as a result of mGlu1 receptor down-regulation. Pharmacological activation of mGlu5 receptors with the selective PAM, VU0360172, improved motor coordination in SCA1 mice, but, as opposed to the mGlu1 receptor PAM, Ro0711401, tolerance to the symptomatic effects of VU0360172 developed [33]. This suggests that mGlu5 receptors in Purkinje cells are not valuable targets for therapeutic intervention in the SCA1 mice model.

### 1.2. Targeting mGlu1 Receptors in the Treatment of Cognitive Dysfunction Associated with SCA1

#### Background

Although motor symptoms are prominent, patients affected by SCA1 may also experience cognitive deficits [4,38,39,40,41,42,43], which may manifest in the early stages of the disease, as shown by patients affected by juvenile-onset SCA1 [44]. SCA1^154Q/2Q^ mice show a defect in learning and memory at 7–8 weeks of age, highlighted by the Morris water maze test, when cerebellar motor symptoms are not yet present [45]. In addition to the cerebellum, the hippocampus is involved in the pathophysiology of learning and memory deficits in the SCA1 mice model [46], and hippocampal pathology (e.g., intranuclear inclusions, abnormalities in dendritic branching, and atrophy) precedes cerebellar pathology in SCA1^154Q/2Q^ mice [45]. mGlu1 receptors are found in different cell types in the hippocampus (e.g., dentate granule neurons, CA2/CA3 pyramidal neurons, and CA1/CA3 interneurons of stratum oriens/alveus) [23,47]. Expression of the α splice variant of mGlu1 receptors is restricted to interneurons [48]. Hippocampal mGlu1 receptors modulate synaptic responses in interneurons, which, in turn, regulate the excitability and synchronization of pyramidal neurons. In addition, mGlu1 receptors are involved in the induction of long-term potentiation (LTP) and contextual learning [9,49,50,51,52,53,54,55,56,57]. Formation of endocannabinoids (i.e., N-arachidonoylethanolamine (AEA or anandamide) and 2-arachidonylglycerol (2-AG)) has been implicated in mechanisms underlying mGlu1 receptor-mediated LTP in the hippocampus. For example, hippocampal stratum oriens somatostatin-positive interneurons undergo endocannabinoid/CB1 receptor-dependent LTP in response to mGlu1 receptor activation [56].

Here, we report that learning and memory impairment in 8-week-old SCA1^154Q/2Q^ mice are associated with reduced expression and function of mGlu1 receptors in the hippocampus and that selective pharmacological activation of mGlu1 receptors enhances AEA formation and corrects the deficit in spatial learning in a SCA1 mice model.

## 2. Materials and Methods

### 2.1. Materials

The selective positive allosteric modulator of mGlu1 receptor Ro0711401 [9H-xanthene-9-carboxylic acid (trifluomethyl-oxazol-2-yl) amide] was kindly provided by LaRoche Ltd., Pharmaceutical Division (Basel, Switzerland). All other chemical products were purchased from Sigma Aldrich (S. Louis, Missouri, MO, USA) and Biorad (Hercules, CA, USA).

### 2.2. Animals

Heterozygous B6.129S-Atxn1tm1Hzo/J mice (stock number 005601) were purchased from The Jackson Laboratory (Bar Harbor, Maine, ME, USA). Wild-type littermates were obtained from the colony. Mice were housed under environmentally controlled conditions (ambient temperature, 22 °C; humidity, 40%) on a 12 h light/dark cycle with water and food ad libitum. All experiments were performed according to the European (86/609/EEC) and Italian (D.Lgs. 26/2014) guidelines of animal care and were approved by the Italian Ministry of Health (authorization n. 35/2021-PR). All efforts were made to minimize animal suffering and the number of animals used. Eight-week-old SCA1 mice were compared with wild-type littermates.

### 2.3. Immunofluorescence Staining

Mice (3–4 per group) were sacrificed, and their brains were fixed in 4% PFA/0.01 M PBS. Cerebellar sections (20 µm) were stained with anti-calbindin antibody (1:500, Abcam, Cambridge, UK). Sections were then incubated for 1 h with secondary donkey anti-rabbit Cy3 antibody (1:200; Jackson Immunoresearch, Cambridge, UK). For Nissl staining in the hippocampus, we used NeuroTrace 500/525 green fluorescence Nissl stain (Invitrogen, MA, USA). Sections were examined with a Zeiss Carl Axiophot2 microscope (Zeiss, Gottingen, Germany) and processed with NIS-elements F3.0.

### 2.4. Open-Field Test

Basal locomotor activity was evaluated in an open-field apparatus. The apparatus consisted of an unexplored cubic box (42 × 42 × 21 cm) with the top left uncovered and transparent plastic walls. The box was attached to an activity monitor supplied with an infrared photobeam interruption sensor, and animal movements were measured and recorded by a computerized analysis system (Open Field Activity System Hardware; Med Associates, Inc., St. Albans, UK). On the test day, mice (6–11 per group) were moved into the testing room and left in their cage for 1 h. Afterwards, mice were placed in the middle of the box floor, where their locomotor activity was recorded for 60 min. The distance travelled was recorded by the software every 5 min for a total of 60 min.

### 2.5. Rotarod Test

Motor performance was assessed using an accelerating Rotarod apparatus (Ugo Basile, Varese, Italy), where mice were forced to manage an accelerating rotarod. The rotarod apparatus is composed of a rotating horizontal cylinder (30 mm) and a motor driver control unit. The cylinder had 5 separate rotating compartments, fully enclosed to ensure that the mice did not jump out of their area. Mice (6–11 per group) were pre-trained on the accelerating rotarod equipped with an automatic fall detector for 3 consecutive trials. The duration of time the mice remained on the rod was recorded by automatic timers, and two infrared beams recorded when the mice had fallen off the rod. The following day, motor performance was calculated as the latency to fall from the accelerating rotating rod from 4 to 30 rpm 3 times.

### 2.6. Morris Water Maze Test

Spatial learning was assessed using the behavioral approach described by Morris [58] with minor modifications. The apparatus consisted of a round water tank (diameter, 97 cm; height, 60 cm) filled with water at 25 ± 1 °C. The platform (diameter, 10 cm) was submerged 1 cm below the water surface and put at the midpoint of one quadrant, in a fixed position, equidistant from the wall and the center of the pool. The apparatus was placed in a room containing different visual cues. Animals (7–13 per group) were submitted to four trials a day, for 4 successive days, during which mice were left in the pool facing the wall and allowed to swim to the reach the platform. If an animal did not find the platform within a period of 60 s, it was gently guided to it. Mice were allowed to stay on the platform for 20 s after reaching it. The starting points varied every time in a pseudorandomized manner. On the day of the test (the fifth day), the probe trial was performed by removing the platform and allowing mice to swim for 60 s in the maze. The time that mice spent in the region near where the platform had been located was recorded by an observer unaware of the treatments. For the evaluation of spatial learning and memory after the treatment with mGlu1 receptors PAM, Ro0711401, mice were treated 1 h prior to each trial and 1 h prior to the probe test.

### 2.7. Western Blot Analysis

Mice (4–8 per group) were sacrificed after the behavioral tests, and the hippocampi and the cerebella were homogenized at 4 °C in RIPA buffer containing a protease inhibitor cocktail and phosphatase inhibitor (PhosSTOP, Roche, Basel, Switzerland). Total protein concentration was assessed by Bradford method with Biorad Protein assay (Biorad, Hercules, CA, USA). Twenty µg of proteins were separated on 8–12% SDS PAGE and transferred onto an immuno-PVDF membrane. Filters were blocked for 1 h in TTBS (100 mM Tris-HCl, 0.9% NaCl, 1% Tween-20 pH 7.5) containing 5% not-fat dry milk. Filters were then incubated with primary antibodies anti-mGlu1α (1:1000, BD Biosciences, San Jose, CA, USA); Homer-1 (1:1000, Millipore, Burlington, MA USA); GluN1 (1:5000, Sigma Aldrich, S. Louis, MO, USA); GluN2A (1:1000, Upstate, Lake Placid, NY, USA); GluN2B (1:1000, Abcam, Cambridge, UK); GluA2 (1:1000, Millipore, Burlington, MA, USA); ph-GluA2 (1:1000, Millipore); CB1R (1:200, Cayman Chemical, Ann Arbor, MI, USA); NAPE-PLD (1:500, Cayman Chemical, Ann Arbor, MI, USA); MAGL (1:500, Abcam); FAAH (1:1000, Santa Cruz, Dallas, TX, USA); and DAGLβ (1:1000, Thermo Fisher, Waltham, MA, USA). After three washes in TTBS, filters were incubated with secondary antibodies HRP conjugated (Millipore) for 1 h at RT. Protein expression was detected by the chemiluminescence (ECL) system, visualized with ChemiDoc XRS (Biorad), and analyzed using ImageJ software (NIH, Bethesda, MD, USA). Filters were then re-incubated with anti-β-actin (1:1000, Sigma Aldrich) or anti-β-tubulin (1:1000, Santa Cruz, Dallas, TX, USA) for protein normalization.

### 2.8. Measurements of Endogenous Levels of Endocannabinoids by LC-MS

The analytical standards that were used were N-arachidonoylethanolamine (AEA), N-arachidonoylethanolamine-d8 (AEA-d8), 2-arachidonoylglicerol (2-AG), and 2-arachidonoylglicerol-d8 (2-AG-d8). The lipid fraction from mouse hippocampi (5–7 per group) was extracted using chloroform–methanol–water (2:1:1 *v/v*) in the presence of ISs (1 ng mL^−1^ of AEA-d8, 100 ng mL^−1^ of 2-AG-d8, and 1 ng mL^−1^ of PEA-d4). The organic phase was dried under a gentle nitrogen stream and then subjected to a micro-solid phase extraction (µSPE) procedure for a rapid clean-up using OMIX C18 tips from Agilent Technologies (Santa Clara, CA, USA). All analyses were performed using a Nexera XR LC 20 AD UHPLC system (Shimadzu Scientific Instruments, Columbia, MD, USA) that was equipped with Kinetex XB-C18 1.7 µm 100 × 2.1 mm from Phenomenex (Torrance, CA, USA) and coupled with a 4500 Qtrap from Sciex (Toronto, ON, Canada) that was equipped with a Turbo V electrospray ionization (ESI) source. The levels of AEA and 2-AG were then calculated as pmoles per mg of tissue.

## 3. Results

As compared to age-matched wild-type mice, 8-week-old SCA1 mice showed no Purkinje cell loss in the cerebellum (Figure 1A), no modifications in mGlu1α receptor protein levels in the cerebellum (Figure 1B), no changes in spontaneous motor activity in an open field apparatus (Figure 1C), and no motor impairment in the rotating rod (Figure 1D), as expected.

However, 8-week-old SCA1 mice showed a significant impairment in spatial learning in the Morris water maze (Figure 1E) and a defect in memory retention, as reflected by a reduction in the time spent in the quadrant where the platform was previously located (Figure 1F).

Eight-week-old SCA1 mice showed a significant reduction in protein levels of mGlu1α receptors and the GluN2B subunit of NMDA receptors in the hippocampus (Figure 2A,C), with no apparent loss of hippocampal neurons (Appendix A). No changes in protein levels of mGlu5 receptors, Homer-1, GluN1 and GluN2A subunits of NMDA receptors, and phosphorylated GluA2 subunit of AMPA receptors were found in the hippocampus of 8-week-old SCA1 mice (Figure 2B,D–F).

To examine whether a reduced expression of mGlu1 receptors could be linked to the impairment in learning and memory, we treated the SCA1 and wild-type mice with either mGlu1 receptor PAM, Ro0711401 (10 mg/kg, s.c.), or its vehicle (sesame oil) every day 1 h prior to each learning session in the water maze and then 1 h prior to the probe test. Selective pharmacological activation of mGlu1 receptors with Ro0711401 corrected the learning and memory deficits in the SCA1 mice model (Figure 3A,B). In wild-type mice, treatment with Ro0711401 did not alter the temporal pattern of learning in the water maze (Figure 3A) but significantly enhanced memory retention in the probe test (Figure 3B).

A great deal of evidence indicates that hippocampal activation of mGlu1 receptors stimulates the formation of endocannabinoids, which, in turn, activates presynaptic CB1 receptors and restrains GABA release from interneurons [56,59,60,61]. Interestingly, CB1 receptor protein expression was strongly reduced in the hippocampus of 8-week-old SCA1 mice (Figure 4).

We therefore examined whether treatment with Ro0711401 caused changes in the levels of the endocannabinoids, AEA and 2-AG, and their main synthesizing and degrading enzymes (NAPE-PLD and FAAH for AEA, and DAGL and MAGL for 2-AG, respectively). Measurements were performed in the hippocampus of the same mice used for the assessment of learning and memory. A five-day treatment with Ro0711401 significantly increased hippocampal AEA levels, leaving 2-AG levels unchanged (Figure 5A,B).

Hippocampal levels of MAGL and DAGL-β did not differ between the two genotypes and were not affected by Ro0711401 treatment (DAGL-α could not be detected with our antibody) (Figure 6A,B). In contrast, hippocampal levels of the AEA synthesizing enzyme, NAPE-PLD, were reduced, and levels of the AEA-degrading enzyme, FAAH, were increased in SCA1 mice treated with Ro0711401, as compared to wild-type mice treated with the vehicle or Ro0711401 (Figure 6A,C).

## 4. Discussion

These findings suggest that hippocampal mGlu1α receptors become defective in the early stages of SCA1, in which cerebellar function is still unimpaired, and that selective pharmacological activation of mGlu1 receptors may improve the defect in learning and memory associated with SCA1. In the hippocampus, mGlu1α receptors are selectively expressed in interneurons [48], as opposed to other mGlu1 receptor splice variants [23]. While hippocampal mGlu1 receptors are implicated in mechanisms of activity-dependent synaptic plasticity involved in learning and memory processes (see Introduction and References therein), the signaling events mediating the overall effects of mGlu1 receptor activation are only partially elucidated. An elegant study has shown that somatostatin-positive interneurons of the hippocampal stratum oriens express both mGlu1 receptors and the enzymatic machinery necessary for endocannabinoid production [56]. Interestingly, we have found that CB1 receptors were down-regulated in the hippocampus of the SCA1 mice model and that mGlu1 receptor activation in the same mice used for the assessment of learning and memory enhanced AEA, but not 2-AG, levels in the hippocampus. To interpret our findings, we adopted the same model proposed by Landucci et al. (this issue of Cells) [62] to explain the protective activity of mGlu1 receptor antagonists against ischemia-induced hippocampal damage [59,62]. It is reasonable that mGlu1 receptors’ endogenous activation promotes synaptic plasticity and learning and memory by regulating the activity of a chain of inhibitory interneurons. Activation of mGlu1 receptors in interneurons would lead to an increased formation of AEA, with ensuing activation of CB-1 receptors and inhibition of GABA release at synapses between interneurons and pyramidal neurons, resulting in the disinhibition of pyramidal neurons. Extrasynaptic NMDA receptors containing the GluN2B subunit might contribute to this mechanism by interacting with mGlu1α receptors through a chain of scaffolding proteins, including PSD-95, Shank and Homer (see Figure 7). According to this model, a down-regulation of mGlu1α, GluN2B, and CB1 receptors nicely explains the learning and memory defect of 8-week-old SCA1 mice. Treatment with Ro0711401 might restore learning and memory in the SCA1 mice model by amplifying the activity of residual mGlu1α receptors leading to increased formation of AEA in the hippocampus. How precisely mGlu1 receptor activation is linked to AEA production is unknown. Unexpectedly, we found a reduction in the levels of the AEA-synthesizing enzyme, NAPE-PLD, and an increase in the levels of the AEA-degrading enzyme, FAAH, in the hippocampus of mice treated with Ro0711401. Although we measured enzyme protein levels and not activity, these changes are not in line with the increased AEA levels found in SCA1 mice treated with Ro0711401. While PLD-mediated hydrolysis of NAPE is the major source of AEA [63], other metabolic pathways have been identified. One pathway involves deacylation of NAPE by α, β-hydrolase-4 and removal of glycerophosphate to generate AEA [64]; the other pathway involves phospholipase-C-mediated hydrolysis of NAPE into phosphoanandamide, which is then converted into AEA by two broad-spectrum phosphatases (the tyrosine phosphatase PTPN22 and the inositol-5′-phosphatase SHIP1) [65]. Both pathways may contribute to AEA biosynthesis in the mouse brain [66]. It is possible that mGlu1 receptors activate the latter alternative pathways through the G_q/11_ mediated coupling with phospholipase-Cβ [23].

## 5. Conclusions

Pharmacological activation of mGlu1 receptors with a subtype-selective PAM represents a potential broad-spectrum strategy in the treatment of SCA1. In the conventional “symptomatic” phase of SCA1, characterized by a progressive cerebellar dysfunction, amplification of mGlu1 receptor signaling in surviving Purkinje cells might improve motor learning and motor coordination, as observed in 30-week-old SCA1 mice [33]. As outlined above, a potential pitfall of this treatment is the induction of excitotoxicity in Purkinje cells made vulnerable by the ongoing pathology. Data obtained in the hippocampus of the “pre-symptomatic” SCA1 mice model also suggest that the early defect in learning and memory associated with SCA1 might be improved by drugs that selectively activate mGlu1 receptors. It is unfortunate that, to our knowledge, there are no mGlu1 receptor PAMs under clinical development, and, therefore, the safety profile of these drugs in humans is unknown. However, optimism is generated by the limited distribution of mGlu1 receptors in the CNS and peripheral organs. mGlu1 receptors have been linked to the pathophysiology of melanoma [67,68]. However, the absence of mGlu1 receptors in melanocytes and benign nevi [67] suggests that treatments with mGlu1 PAMs can be harmful only if melanoma cells are already present.

## Figures and Tables

**Figure 1 cells-11-03916-f001:**
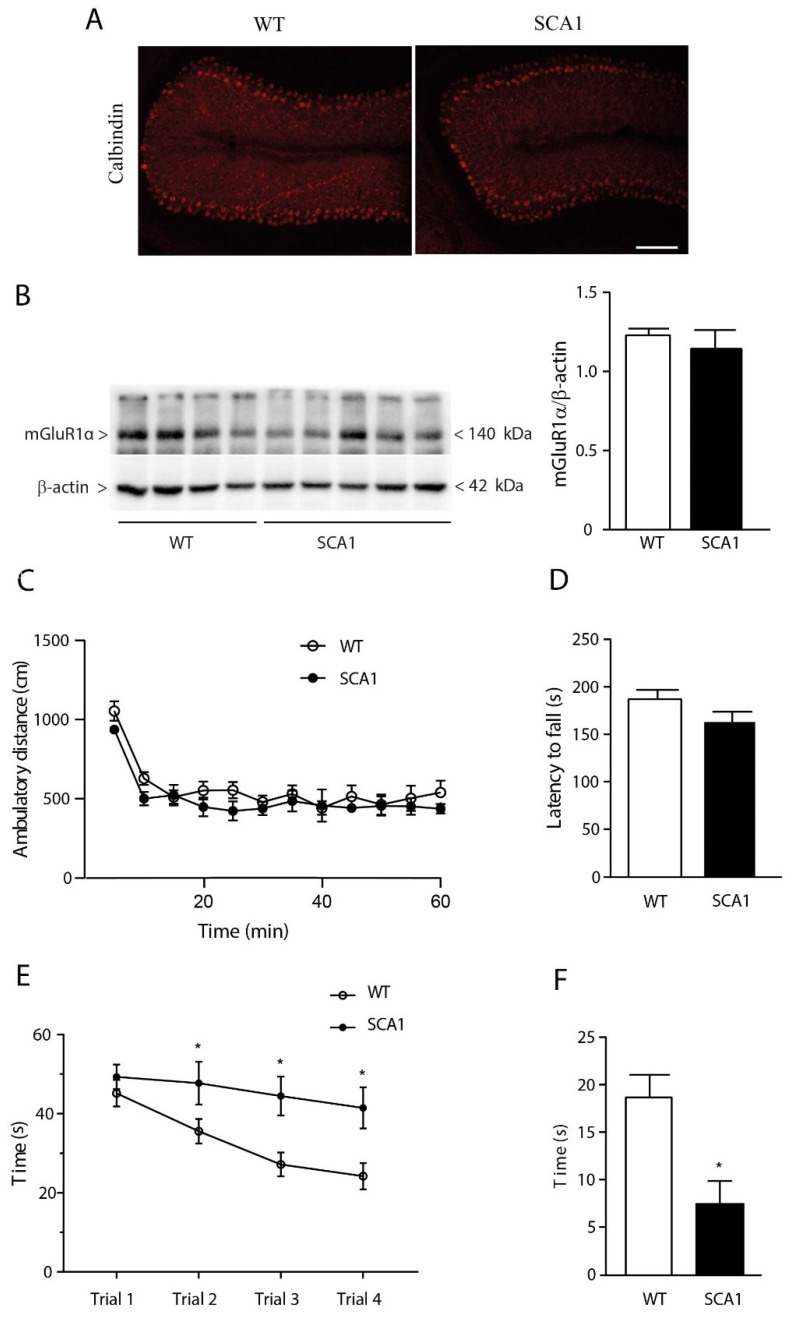
Impaired spatial learning and memory in 8-week-old SCA1 mice showing no cerebellar abnormalities. Calbindin immunostaining of cerebellar Purkinje cells in 8-week-old SCA1 mice and wild-type littermates is shown in (**A**). Scale bar: 100 µm. Immunoblot analysis of mGlu1α receptors in the cerebellum is shown in (**B**), where values are means ± S.E.M. of 4–5 mice; motor performance in the open-field apparatus is shown in (**C**), where values are means ± S.E.M. of 6–11 mice; motor coordination in the rotating rod is shown in (**D**), where values are means ± S.E.M. of 6–11 mice; spatial learning and memory retention in the two genotypes are shown in (**E)** and (**F**), respectively, where values are means ± S.E.M. of 7–13 mice. * *p* < 0.05 vs. the respective values obtained in wild-type mice. Statistical analysis was performed by two-way ANOVA for repeated measures + Bonferroni test in (**E**) and Student’s *t*-test in (**F**). In (**E**): genotype, F_1,18_ = 9.573, *p* = 0.0063; time, F_3,54_ = 8.648, *p* < 0.0001; interaction, F_3,54_ = 2.035, *p* = 0.1198; subject, F_18,54_ = 3.565, *p* = 0.0002. In (F), t_18_ = 3.139.

**Figure 2 cells-11-03916-f002:**
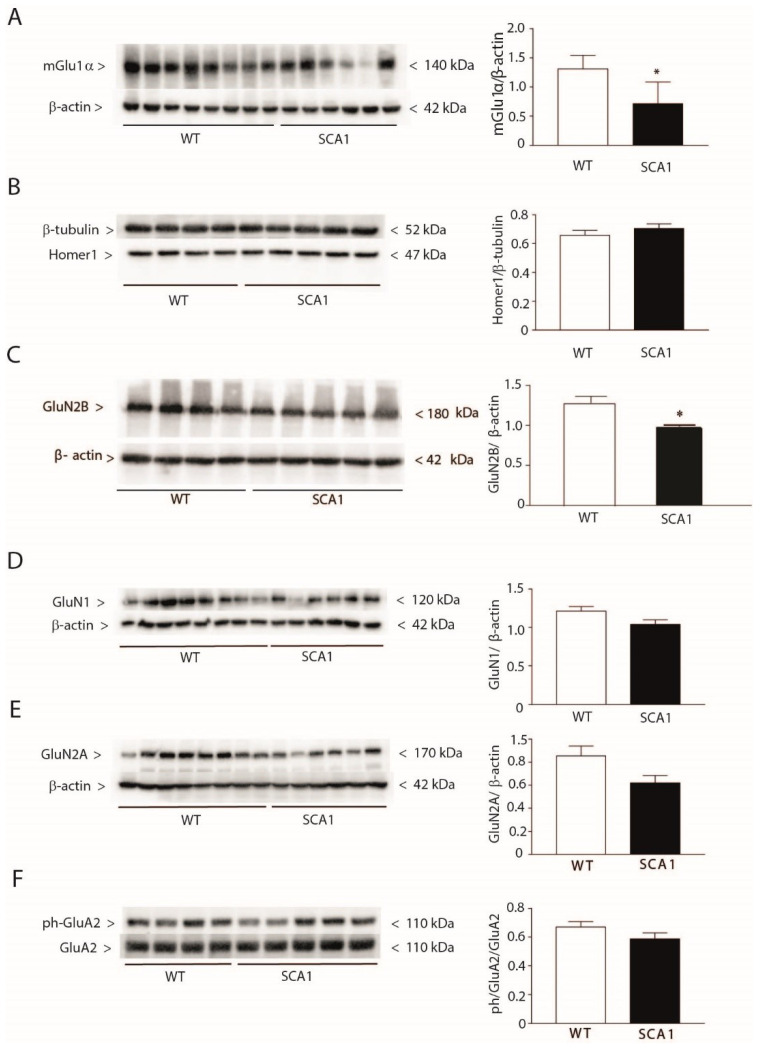
Reduced mGlu1α and GluN2B protein levels in the hippocampus of 8-week-old SCA1 mice. Hippocampal protein levels of mGlu1α receptors, pan-Homer1, GluN2B, GluN1 and GluN2A subunits of NMDA receptors, and phospho-GluA2 subunit of AMPA receptors of 8-week-old SCA1 mice and wild-type littermates are shown in (**A**), (**B**), (**C**), (**D**), (**E**), and (**F**), respectively. Values are means ± S.E.M. of 6–8 (**A**), 4–5 (**B,C,F**), and 6–8 (**D,E**) mice. * *p* < 0.05 vs. wild-type mice (Student’s *t*-test). In (**A**), t_12_ = 3.709; in (**C**), t_7_ = 3.892.

**Figure 3 cells-11-03916-f003:**
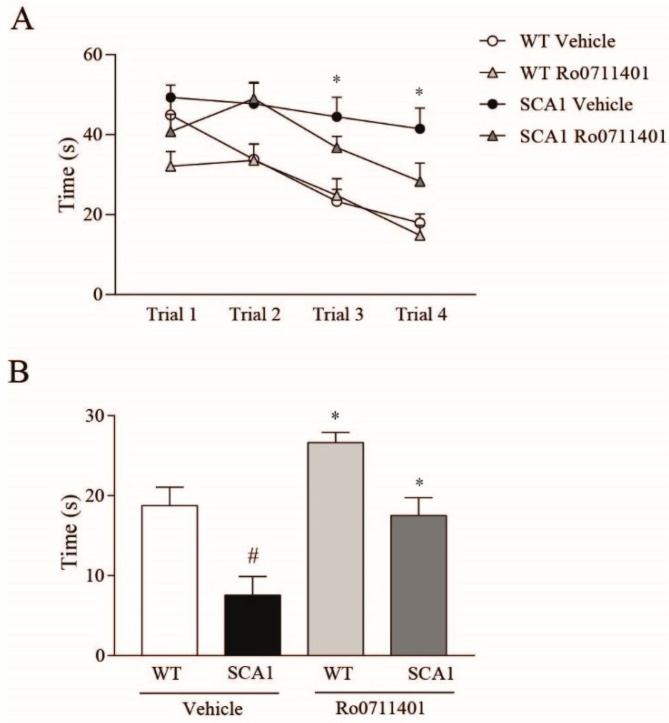
Treatment with the mGlu1 receptor PAM, Ro0711401, improves learning and memory in 8-week-old SCA1 mice. Spatial learning and memory retention in 8-week-old SCA1 mice and wild-type littermates are shown in (**A**) and (**B**), respectively. * *p* < 0.05 SCA1 vehicle vs. the respective values obtained in wild-type mice in (**A**). * *p* < 0.05 vs. SCA1 vehicle; # *p* < 0.05 vs. the respective values obtained in WT mice treated with vehicle in (**B**). Statistical analysis was performed by two-way ANOVA for repeated measures + Bonferroni test in (**A**) and two-way ANOVA + Bonferroni test in (**B**). Values are means ± SEM of 7–10 mice. In (**A**): interaction, F_9,93_ = 2.71, *p* = 0.0076; time, F_3,93_ = 30.19, *p* < 0.0001; treatment, F_3,31_ = 7.474, *p* = 0.0007; subject, F_31,93_ = 4.545, *p* < 0.0001. In (**B**): genotype, F_1,30_ = 24.02, *p* < 0.0001; treatment, F_1,30_ = 15.29, *p* = 0.0005; interaction, F_1,30_ = 0.9748, *p* = 0.3314.

**Figure 4 cells-11-03916-f004:**
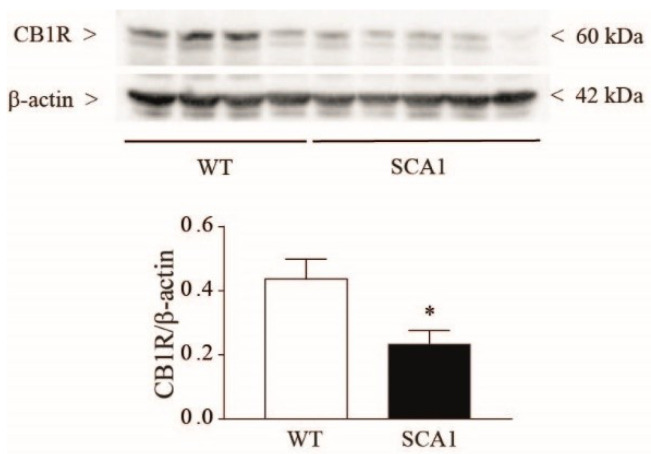
Reduced CB1R protein levels in the hippocampus of 8-week-old SCA1 mice. Hippocampal protein levels of CB1 receptors. Values are means ± S.E.M. of 4–5 mice. * *p* < 0.05 vs. wild-type mice (Student’s *t*-test); t_7_ = 2.413.

**Figure 5 cells-11-03916-f005:**
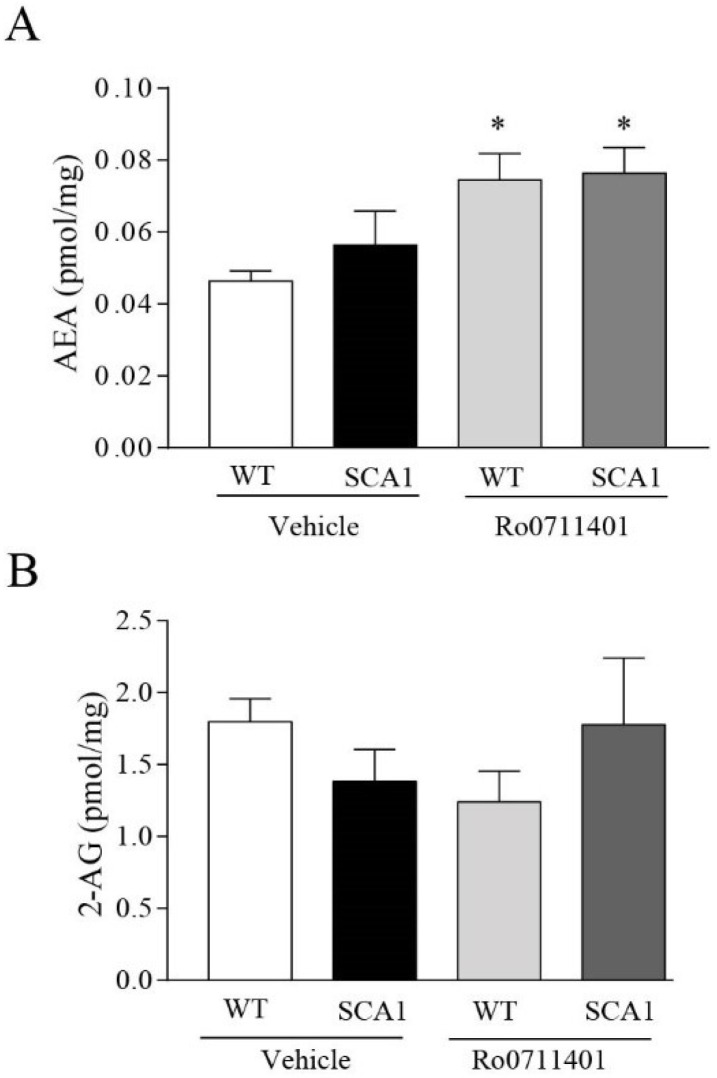
Increased hippocampal levels of AEA in 8-week-old SCA1 mice and their wild-type littermate treated with Ro0711401. Increased AEA hippocampal levels in the SCA1 mice model and wild-type littermates treated with Ro0711401 are shown in (**A**), while no changes in hippocampal 2-AG levels are shown in (**B**). * *p* < 0.05 vs. WT vehicle. Statistical analysis was performed by two-way ANOVA + Bonferroni test. Genotype, F_1,18_ = 0.7344, *p* = 0.4027; treatment, F_1,18_ = 12, *p* = 0.0028; interaction, F_1,18_ = 0.3286, *p* = 0.5735. Values are means ± SEM of 5–6 (**A**) and 6–7 (**B**) mice.

**Figure 6 cells-11-03916-f006:**
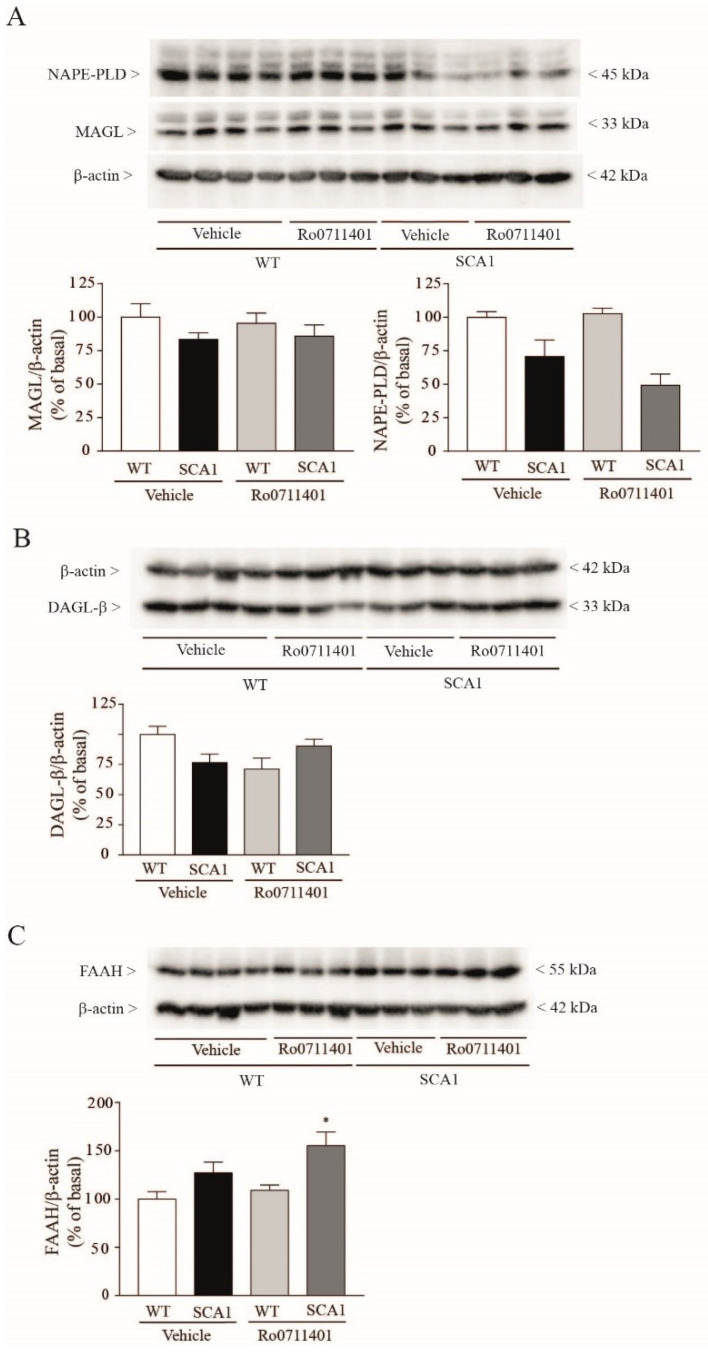
Changes in hippocampal protein levels of NAPE-PLD and FAAH in SCA1 mice treated with Ro0711401. Reduced hippocampal levels of the AEA synthesizing enzyme, NAPE-PLD, in SCA1 mice treated with Ro0711401 are shown in (**A**), while increased levels of the AEA-degrading enzyme, FAAH, are shown in (**C**), as compared to wild-type mice treated with vehicle or Ro0711401. Hippocampal levels of MAGL and DAGL-β did not differ between the two genotypes and were not affected by Ro0711401 treatment (**A, B**). * *p* < 0.05 vs. the respective values obtained in wild-type mice. Statistical analysis was performed by two-way ANOVA + Bonferroni test values are means ± SEM of 4–5 mice. In (**A**): genotype, F1,15 = 24.01, *p* = 0.0002; treatment, F_1,15_ = 1.216, *p* = 0.2876; interaction, F_1,15_ = 2.038, *p* = 0.1739. In (**C**): genotype, F_1,15_ = 12.2, *p* = 0.0033; treatment, F_1,15_ = 3.12, *p* = 0.0977; interaction, F_1,15_ = 0.8092, *p* = 0.3826.

**Figure 7 cells-11-03916-f007:**
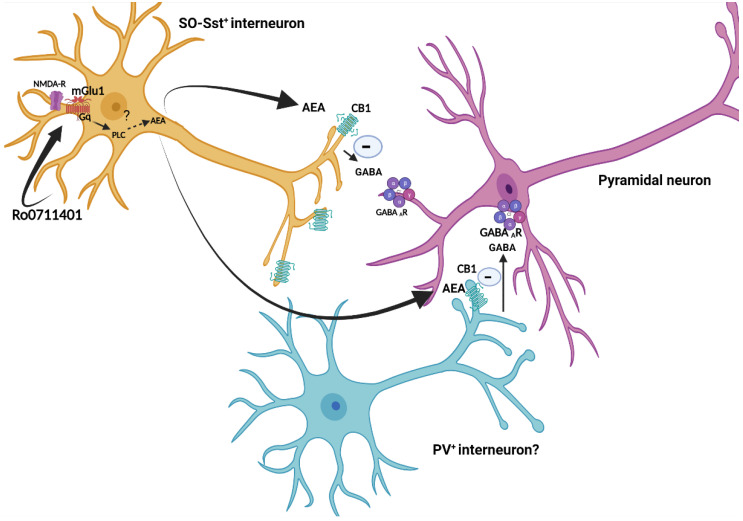
Hypothetical mechanism of Ro0711401-induced learning and memory improvement in SCA1 mice model. Selective pharmacological activation of mGlu1 receptors in stratum oriens somatostatin-positive interneurons (SO-sst^+^ interneurons) enhances anandamide production through a non-conventional synthetic pathway that may involve G_q_-mediated PLC-β activation. Activation of CB1 receptors in nerve terminals of somatostatin-positive interneurons or other interneurons causes inhibition of GABA release with ensuing activation of pyramidal neurons. A similar scenario has been proposed by Landucci et al., 2022 [62]. Created with BioRender.com.

## Data Availability

All raw data will be subsequently deposited.

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
