# Peer review of "Targeting mGlu1 Receptors in the Treatment of Motor and Cognitive Dysfunctions in Mice Modeling Type 1 Spinocerebellar Ataxia"

_cells, 2022, doi:10.3390/cells11233916_

Round 1

Reviewer 1 Report

The manuscript of Liberatore et al. reports that mGlu1 receptor activation with a drug Ro0711401 may be beneficial in restoring non-motor symptoms in SCA1 8-week mice model. Additionally, the authors analyze molecular pathways and other receptors that may be involved in this process. Although a similar study has been performed for older mice, it is important to show that the treatment is beneficial also in the early disease stages.

The introduction is well-written and the molecular part is clearly explained. The methods are well described and the methodology is sound. Results are clearly presented and the discussion provides a possible explanation. Conclusions could be more concise, but they include the most important aspects.

Comments:

1. Image A Figure 1: Hardy anything is seen. Maybe it would help to make a picture larger or in better quality?

2. It the introduction it could be mentioned that the same therapy improved ataxia in older SCA1 model mice https://doi.org/10.1186/1756-6606-6-48

3. In the methods I would state clearly how many mice were included in each experimental part.

4. Instead of "SCA1 mice" I would use "SCA1 mice model" or "SCA1 transgenic mice model"

5. Line 148-149 I would briefly mention tests based on which the cognitive dysfunction has been reported in mice

6. Line 185: I would use "sacrificed" instead of "killed".

7. Lines 417-423: I would include this part in the discussion, not in the conclusions

8. Line 434: Please make the acknowledgment more precise.

Author Response

Dear Reviewer 1, thank you for your suggestions, we have made all the changes you requested:

  1. Image A Figure 1: Hardy anything is seen. Maybe it would help to make a picture larger or in better quality?

Figure 1 has been enlarged

  1. It the introduction it could be mentioned that the same therapy improved ataxia in older SCA1 model mice https://doi.org/10.1186/1756-6606-6-48

The reference about the therapy in older SCA1 model mice has been added at lines 105 and 113

  1. In the methods I would state clearly how many mice were included in each experimental part.

The number of the animals used has been added in the methods section

  1. Instead of "SCA1 mice" I would use "SCA1 mice model" or "SCA1 transgenic mice model"

“SCA1 mice” has been replaced by “SCA1 mice model”

  1. Line 148-149 I would briefly mention tests based on which the cognitive dysfunction has been reported in mice

The test used to evaluate cognitive dysfunction has been added at line 150

  1. Line 185: I would use "sacrificed" instead of "killed".

The term “killed” has been replaced by “sacrificed” at lines 188 and 234

  1. Lines 417-423: I would include this part in the discussion, not in the conclusions

Lines 417-423 are part of the Figure legend of the Figure 7.

  1. Line 434: Please make the acknowledgment more precise.

Acknowledgment has been modified.

Finally, in order to reduce the duplicate rate, we accepted the Assistant Editor’s suggestion and we highlighted the modified parts in yellow.

With my best regards,

Sincerely,

Serena Notartomaso

Reviewer 2 Report

The authors claim that mGlu1 receptor activation improves some cognitive and motor deficits of 6.129S-Atxn1tm1Hzo/J mice at age 8 weeks. The presentation is professional and knowledgable, the findings are interesting and worth reporting.

Major criticism:
The data seem valid for a specific age, a specific animal model, and specific test parameters. As in any association study, after this hypothesis-generating tier it would be desirable to have confirmation of the claim either in another SCA1 model, or at least at another age, otherwise all effects can simply be due to multiple testing of many phenotype parameters and many molecules.

Author Response

Dear Reviewer 2, thank you for your precious suggestions.

Major criticism:
The data seem valid for a specific age, a specific animal model, and specific test parameters. As in any association study, after this hypothesis-generating tier it would be desirable to have confirmation of the claim either in another SCA1 model, or at least at another age, otherwise all effects can simply be due to multiple testing of many phenotype parameters and many molecules.

We completely agree. These findings are promising from a therapeutic standpoint pending confirmation in other mouse models of SCA1. This will be addressed in an entirely new project that requires several months for local and ministerial approval before we can perform additional experiments. The request to extend the findings to other ages in the current SCA1 model is also relevant. However, it is difficult to study spatial learning and memory with the Morris water maze in older SCA1 mice because of their ataxic phenotype. We should therefore search for other tests that are minimally influenced by motor performance. This will be developed in the future.

Finally, in order to reduce the duplicate rate, we accepted the Assistant Editor’s suggestion and we highlighted the modified parts in yellow.

With my best regards,

Sincerely,

Serena Notartomaso